# A CNN-Based Strategy to Classify MRI-Based Brain Tumors Using Deep Convolutional Network

**Ahmed Wasif Reza** [1], **Muhammad Sazzad Hossain** [1], **Moonwar Al Wardiful** [1], **Maisha Farzana** [1], **Sabrina Ahmad** [1], **Farhana Alam** [1], **Rabindra Nath Nandi** [2] **and Nazmul Siddique** [3,*]

1    Department of Computer Science and Engineering, East West University, Dhaka 1212, Bangladesh
2    Department of Artificial Intelligence, BJIT Limited, Dhaka 1212, Bangladesh
3    School of Computing, Engineering and Intelligent Systems, Ulster University, Northland Road, Londonderry BT48 7JL, UK
*    Correspondence: nh.siddique@ulster.ac.uk

**Abstract:** Brain tumor is a severe health condition that kills many lives every year, and several of those casualties are from rural areas. However, the technology to diagnose brain tumors at an early stage is not as efficient as expected. Therefore, we sought to create a reliable system that can help medical professionals to identify brain tumors. Although several studies are being conducted on this issue, we attempted to establish a much more efficient and error-free classification method, which is trained with a comparatively substantial number of real datasets rather than augmented data. Using a modified VGG-16 (Visual Geometry Group) architecture on 10,153 MRI (Magnetic Resonance Imaging) images with 3 different classes (Glioma, Meningioma, and Pituitary), the network performs significantly well. It achieved a precision of 99.4% for Glioma, 96.7% for Meningioma, and 100% for Pituitary, with an overall accuracy of 99.5%. It also attained better results than several other existing CNN architectures and state-of-the-art work.

**Keywords:** brain tumor; MRI; multiclass classification; deep learning; VGG; CNN

## 1. Introduction

A brain tumor is a mass developed by abnormal cell growth and division inside the skull. Brain tumors are rare and can be fatal [1]. They come in a variety of shapes and sizes and can arise in any place and with varying image intensities [2]. Depending on their origin, brain tumors are classified as primary or metastatic. Primary cancer cells originate in brain tissue [3], whereas metastatic cancer cells become malignant in any other part of the body and spread to the brain [1]. A timely diagnosis of a brain tumor is critical for optimal treatment planning and patient care.

Various medical imaging techniques are applied to collect information about tumors. Imperative innovations incorporate Computed Tomography (CT), positron emission tomography (PET), Single-Photon-Emission Computed Tomography (SPECT), Magnetic Resonance Spectroscopy (MRS), and Magnetic Resonance Imaging (MRI). These technologies can be used in conjunction to gather more specific information about tumors. MRI, on the contrary, is the most commonly employed method because of its beneficial properties. MRI is a non-invasive in vivo imaging method that uses radiofrequency waves to trigger target tissues, causing them to form internal images under the influence of a superconducting magnet [4]. The scan delivers hundreds of 2D image slices with good soft tissue contrast in MRI collection, while not using ionizing radiation [5,6]. During image acquisition, excitation and repetition periods are modulated to create images of varied MRI sequences.

The radiologists rely on their training and experiences to manually determine the abnormality of the brain MRI, then categorize them into tumor types [7]. Early detection and classification of brain tumors play a key role in the evaluation case and, consequently,

contribute to the selection of the most appropriate treatment to save patients' lives [8]. One of the major difficulties of manual detection is the chance of misclassification of the tumor, which can lead to the wrong treatments for the patients. In addition, as time plays an important role, manual detection has no major advantage here. Therefore, the desire for an automated and quick detection technique is expected [9].

The classification of brain tumors into subgroups is a more difficult scientific problem. The factors causing the problems are brain tumors that vary widely in form, size, and intensity [10], and tumors of different pathological categories that may seem identical [11]. We are keen on classifying abnormal types and normal brain images in this study. The MRI dataset used in this study includes images of the brain without tumors, as well as three different forms of brain tumors. Glioma, meningioma, and pituitary tumors account for approximately 45%, 15%, and 15% of all brain tumors, respectively, in clinical practice [12]. This work is more sophisticated and demanding than conventional binary classification (normal and abnormal), as not only can it identify the problem, it also has the ability to categorize the abnormalities. Machine learning techniques are now frequently utilized in medical imaging [6]. To estimate new topic labels in supervised techniques, an algorithm is used to find a mapping function of input variables and their related output labels. The primary goal is to find intrinsic patterns in training data using techniques, such as Artificial Neural Network (ANN) [13], Support Vector Machine (SVM), and K-Nearest Neighbors (KNN) [14]. Unsupervised learning, on the other hand, is based solely on input variables, as shown by fuzzy c-means [15] and the Self-Organization Map (SOM) [16]. To establish learning, the features of the training images must be extracted, which are typically grayscale, texture, and statistical characteristics. These characteristics are called handmade features and they require the expertise of a specialist with considerable knowledge and the ability to select the most vital aspects. Furthermore, this operation takes a long time and is prone to errors when dealing with large amounts of data [17].

Deep learning was recently introduced to the medical imaging area and has shown substantial success in classification problems, specifically the multiclass classification problem with better accuracy [18–22]. Deep learning algorithms use a matrix of multiple layers of asymmetric processing techniques to extract features.

Convolutional Neural Network (CNN) comprises several convolutional layers, pooling layers, and fully connected layers for segmentation and feature extraction, reduction of the spatial size of the representation, and classification [23,24]. The frequently used activation functions in CNN include ReLU (Rectified Linear Unit) [25], FReLU [26], LeakyReLU [27], Swish [28], ACON (Activate or Not) [29], and SoftMax [30].

In this study, our goal was to create an artificial model which can predict the types of tumors accurately within a few seconds. The proposed model is a modified CNN model, inspired by the Visual Geometry Group (VGG) architecture, which performs significantly well than many other architectures. Working with medical images is harder than usual, because of the sensitivity of the results, as a wrong diagnosis can result in a life-threatening condition. Therefore, our focus was primarily on getting a high accuracy that can exceed the results of other commonly used architectures.

After the introduction, we discuss related works regarding our study and their research analogy in the Related Works section. Training of the model, along with performance measurements presented in the Materials and Methods section. We have analyzed the result of our proposed architecture in the Results section, discussed the findings and compared its performance with other methods in the Discussion section. Finally, we conclude the paper in the Conclusions section.

## 2. Related Works

Various approaches have been developed in recent years to recognize brain tumors on MRI images. SVM and Neural Networks (NN) are the most widely used techniques due to their excellent performance over the past decade [31].

A Probabilistic Neural Network (PNN) is used where the decision-making strategy partitioned into extraction utilizing vital component investigation and classification using PNN, with an accuracy ranging from 100% to 73% depending on the spread value [32].

A CNN-based deep learning model is proposed for categorizing various forms of brain tumors, where the architecture has a classification accuracy of 96.13% for the categorization of brain tumor types [33].

Extreme Learning Machine of Local Receptive Fields (ELM-LRF) is a new deep learning paradigm that covers two distinct structures in its body [31]. An ELM-LRF model is created with four adjustable parameters, the convolution filter size r, the number of convolution filters K, the pooling size, and the regulatory coefficient C. The proposed ELM-LRF approach yielded a classification accuracy of 97.18% [34].

A Deep Neural Network (DNN) is used as a discriminator in a Generative Adversarial Network (GAN) to extract powerful features and grasp the structure of MRI images in its convolutional layers, resulting in 95.6% accuracy [35].

Multiple Kernel-Based Probabilistic Clustering (MKPC) is used to segment the image, and a deep learning classifier is used to categorize it, achieving an accuracy of 0.83% [36].

A CNN technique is proposed, where the Fuzzy C-Means (FCM) method is used for brain segmentation and texture and form properties from these segmented areas were recovered before being sent into the SVM and DNN processors, with an accuracy of 97.4% [37].

A tiny kernel CNN model can also be used for the classification with $3 \times 3$ kernels for all convolutional layers with 1 stride. This result continues to demonstrate a 90.67% precision in the augmented dataset [38].

In a study, a DNN classifier combined with the Discrete Wavelet Transform (DWT) achieved a good result of 96.77% in a tiny dataset [39].

A deep learning technique is provided to classify multimodal brain tumors using a linear contrast augmentation approach. Before fusion, features are extracted using transfer learning from two distinct CNN models. ELM is used to classify the robust properties obtained by this technique [40].

Several researchers employed pre-trained CNN architectures and fine-tuned them for brain tumor classification. The proposed categorization system extracts attributes from brain MRI images using a pre-trained GoogLeNet. After that, by using proven classifier models and a five-fold cross-validation technique, the experiment obtains 97% accuracy [41].

The Grab cut method is used to properly segment the real lesion [17]. This study shows segmentation utilizing Unet architecture with ResNet50 as a baseline. The application of evolutionary methods (ResNet50, DenseNet201, MobileNet V2, and InceptionV3) and reinforcement learning through transfer learning achieves an accuracy of 91.8%, 92.8%, 92.9%, 93.1%, and 99.6%, respectively, in the categorization of brain cancers.

A multi-level CNN model is introduced, where pre-trained models, such as ResNet-50, VGG-16, and Inception V3, are used to generate trained parameters, achieving 99.89% classification accuracy [42].

A hybrid deep learning model called DeepTumorNet is proposed, which is generated by modifying the layers of GoogleNet architecture, with the addition of leaky ReLU activation function. This architecture achieved 99.67% accuracy [43].

A differential deep convolutional neural network model is suggested, where the differential feature maps of CNN are derived using differential operators, which resulted in an accuracy of 99.25% [44].

Some of the other state-of-the-art work are shown in Table 1.

**Table 1.** Some other state-of-the-art work.

| Author(s) | Concept | Method | Findings | Gaps |
|---|---|---|---|---|
| Deepak, Ameer [41] | Designed a model to classify three pathological types of brain tumor. | Using deep transfer learning and a pre-trained GoogLeNet to extract features, a classifier to classify the types. | For a small dataset, higher classification accuracy was observed. | Higher misclassification in the confusion matrix, overfitting because of a small dataset. |
| Emrah Irmak [45] | Three types of classification tasks have been performed. | Three CNN models perform three classification tasks, in which hyperparameters have been manually optimized using a grid. | Using the grid optimizer is effective as it could find the best model for classification types. | Three classification systems for all three types, a joint multi-classification system can decline its necessity. |
| Sharif, Attique, Musaed, Khursheed, Mudassar [18] | Brain tumor classifications on four types of MRI images, such as T1W, T1CE, T2W, and Flair. | Selection of the most optimal features using Modified Genetic Algorithm (MGA) and entropy-Kurtosis-based techniques and trained by a fine-tuned pre-trained DenseNet201 | Using a feature selection technique improved the result of a publicly available dataset. | Reducing certain key features could have a great impact, as it could help the system achieve accuracy. |

## 3. Materials and Methods

Figure 1 shows the workflow of our proposed model. First, images are loaded to go through several crucial pre-processing stages. Then, the dataset is split into two parts: training and testing.

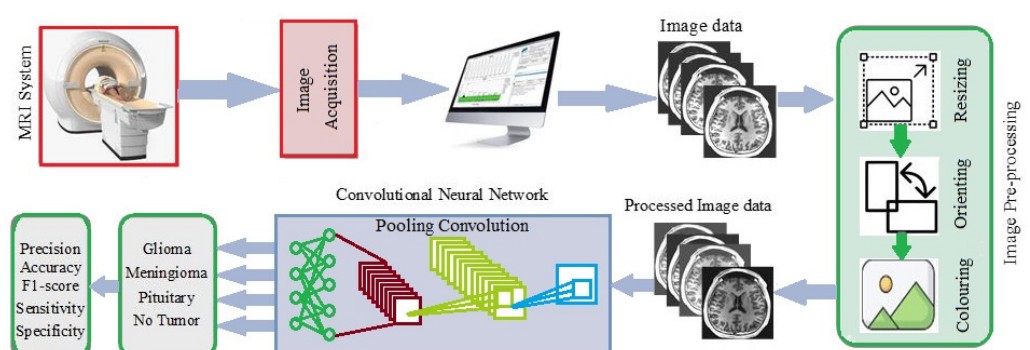

**Figure 1.** A high-level workflow diagram of the proposed method.

### 3.1. Dataset

For this study, we were looking for a comparatively large dataset as we wanted to avoid the data augmentation method. Furthermore, we needed a dataset that contained normal and abnormal brain images and their subtypes. To address this problem, we combined two publicly available Kaggle datasets [46,47]. We added a small amount of data from various sources to enlarge the dataset. Finally, a dataset of 10,153 MRI images was obtained for which the samples from each class are shown in Figure 2. The number of images are 10,153. The number of data from each class are 2547, 2582, 2658, and 2396 for Glioma, Meningioma, Pituitary, and No Tumor, respectively. The data ranges from 2.3 k to 2.6 k. There is no major issue with data imbalance within the dataset, therefore we did not have to implement any kind of extra techniques to handle this insignificant data imbalance.

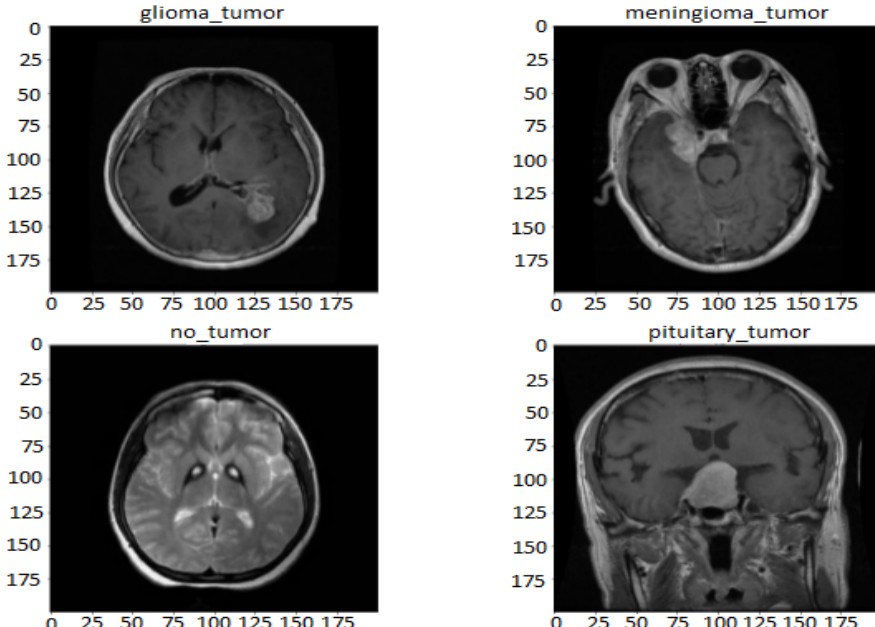

**Figure 2.** Images of the normal brain and the other three types of brain tumors—Glioma, Meningioma and Pituitary.

### 3.2. Preprocessing

The images in the dataset needed to go through some pre-processing stages before training the proposed model. The images were of different sizes. To train our model and get the best accuracy, we resized the image into $200 \times 200 \times 1$ pixels. As a result, it gives better performance and straightforward calculation. We choose to decrease the MRI picture size from an average of $256 \times 256 \times 1$ (the highest being at the size of $500 \times 500 \times 1$) to $200 \times 200 \times 1$ due to the computational restrictions of fitting the complete image to the model here. The required size is chosen so that all parts of the skull are captured, and the images have a centering effect after cropping and resizing. We used the same orientation of the raw data. We have used grayscale images, and to ensure that no images contained any unwanted colors, we have turned them into grayscale images before proceeding any further. Before splitting the data set, we shuffled the dataset and split it into two parts, training and testing, where the training dataset is 80% and the testing dataset is 20%. The dataset is divided here into 80% for training and 20% for testing. We did not perform any data validation separately, however, we have used that 20% testing data as validation during training.

### 3.3. Proposed Model

Figure 3 shows the architecture of the proposed model. Here, we have implemented a similar structure to the VGG-16 architecture. VGG-16 is a deep CNN architecture that contains numerous different layers. The VGG model inspired us to utilize several (deeper architecture) convolution layers to use a restricted receptive field, followed by a max pooling layer to decrease image dimensionality by decreasing the number of pixels in the convolution layer output.

CNN is a very deep architecture for a large number of image datasets for image recognition [48]. It gives a particularly good accuracy in large-scale image processing. In our model, there are 21 layers, where the first 20 layers use the ReLU (Rectified Linear Unit) function, and the last one is the SoftMax function.

Overfitting is a common problem in machine learning while training a large model with a high number of parameters and a relatively small training dataset. Dropout regularization is a technique for combating overfitting. We have used dropout layers to reduce overfitting. The moving edges of hidden node neurons that make up the hidden layers

are randomly set at 0 while updating the training phase [35]. In the proposed model, we found that 20% dropout gives the most accurate values. In Figure 4, all nodes relate to the output layer. After dropping out, some of the nodes are avoided to help our model to reduce overfitting.

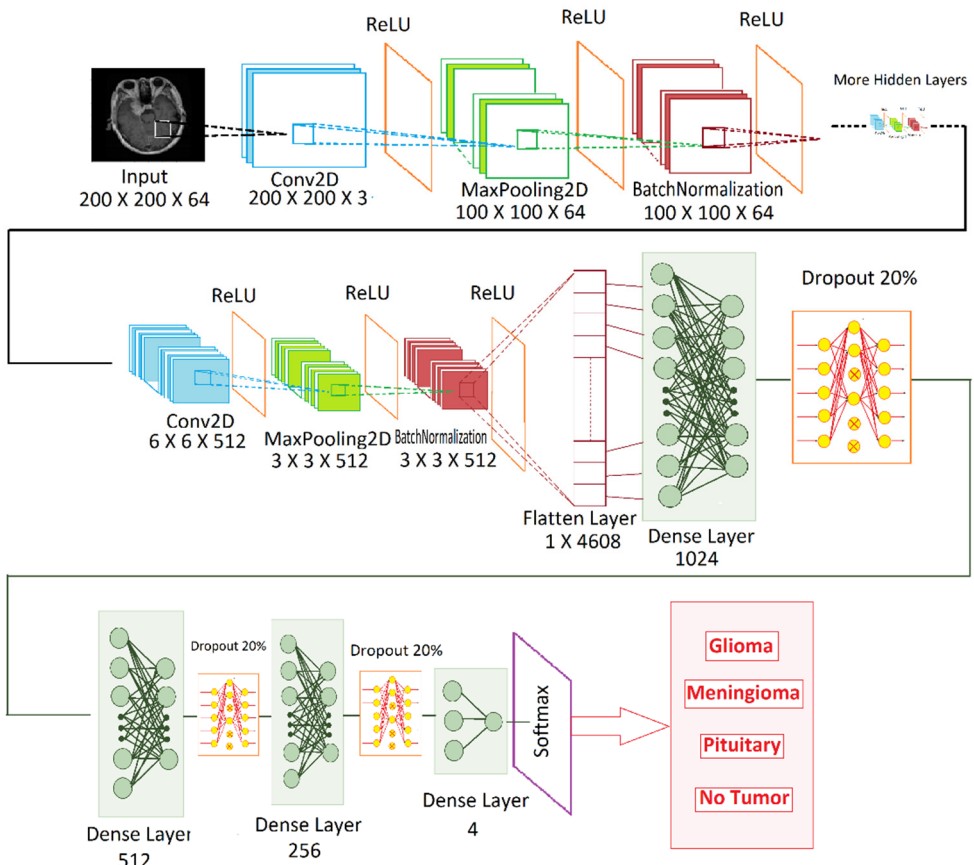

**Figure 3.** The modified VGG-16 model.

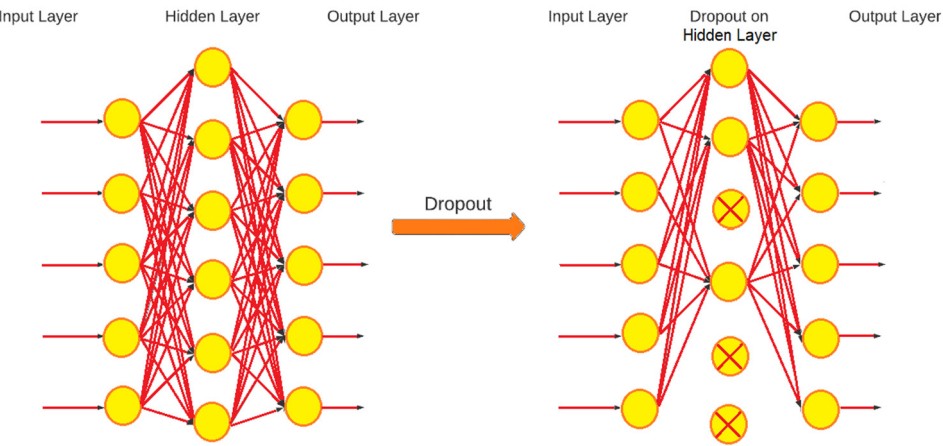

**Figure 4.** Comparison between before and after dropout of nodes.

In our proposed model, we have used a Stochastic Gradient Descent (SGD) optimizer. SGD subtracts the weights from the gradient multiplied by the learning rate. SGD has strong theoretical foundations and is still used in edge-training NNs, despite its simplicity [49].

$$\theta_i = \theta_i - \alpha \frac{\partial L}{\partial \theta_i} \tag{1}$$

Batch normalization is an effective CNN training technique that processes the input to each layer for each mini-batch [50]. The function of batch normalization used here is

$$y_i = \gamma_B.\overline{x_i} + \beta_B \tag{2}$$

In the proposed model, the last layer is softmax.

After finding the normalization in channel B for a mini-batch, an arbitrary value is multiplied, which works as a scaling transformation, and another arbitrary value is added to shift it. These values are set by 0/1 as default and get updated after each epoch. This function reduces the problem with input value changing by stabilizing the values, thus increasing the training speed in our model. We divide our batch into 42 images.

To build our model, we used different parameters and different layers. The summary of the model is given in Table 2.

**Table 2.** The layers of the proposed model.

| Layers | Output Size | Parameters |
| --- | --- | --- |
| Conv2D | None,200,200,3 | 1664 |
| MaxPooling 2D | None,100,100,64 | 0 |
| BatchNormalization | None,100,100,64 | 256 |
| Conv2D | None,100,100,128 | 204,928 |
| MaxPooling 2D | None,50,50,128 | 0 |
| BatchNormalization | None,50,50,128 | 512 |
| Conv2D | None,50,50,128 | 409,728 |
| MaxPooling 2D | None,25,25,128 | 0 |
| BatchNormalization | None,25,25,128 | 512 |
| Conv2D | None,25,25,256 | 819,456 |
| MaxPooling 2D | None,12,12,256 | 0 |
| BatchNormalization | None,12,12,256 | 1024 |
| Convo2D | None,12,12,256 | 1,638,656 |
| MaxPooling | None,6,6,256 | 0 |
| BatchNormalization | None,6,6,256 | 1024 |
| Convo2D | None,6,6,512 | 3,277,312 |
| Maxpooling | None,3,3,512 | 0 |
| BatchNormalization | None,3,3,512 | 2048 |
| Flatten | None, 4608 | 0 |
| Dense layer | None, 1024 | 4,719,616 |
| Dropout 20% | None, 1024 | 0 |
| Dense layer | None, 512 | 524,800 |
| Dropout 20% | None, 512 | 0 |
| Dense layer | None, 256 | 131,328 |
| Dropout 20% | None, 256 | 0 |
| Dense layer | None, 4 | 1024 |
| Softmax | None,4 | 0 |

After the input layer, there are six sets of layers in one set, which are Convo2D, MaxPool, and batch normalization layers with different shapes. Then, there is a flatten layer. After that, there are three dropout layers and three dense layers. In this case, it is a multi-class.

*3.4. Pseudocode*

Input:

       Xt: Brain tumour pre-processed train dataset;

       Xv: Brain tumour pre-processed test dataset;

       $\varepsilon$: Number of epochs;

       $\eta$: Learning Rate;

       B: Batch Size;

Output:

       Assessment Metrics (accuracy etc.) calculation on test dataset.

Start Procedure

       Add_Conv2D (filters, kernel_size, padding, activation)

       Add_MaxPool2D (pool_size)

       Add_BatchNormalization ()

       Add_Flatten ()

       Add_Dense ()

       Add_Dropout (0.2)

       Optimizing with Stochastic gradient descent ($\eta$)

       for all epochs in 1 to $\varepsilon$ do

           for B $\in$ a random batch from Xt do

               model_fit with test data (Xv)

               append (Accuracy)

           Endfor

       Endfor

       Evaluate trained Model dataset -> totalAccuracy

       return totalAccuracy

EndProcedure

## 4. Results

We have used modified VGGNet-CNN architecture for the proposed system. In total, 10,153 samples of MRI images with an input vector size of $200 \times 200$ were used. The dataset was partitioned into two sets, 80% training set and 20% testing set. For validation, 80% of the data from the testing set were kept, and the rest were for the actual testing. We tested the model with three epoch settings (10, 20, 30) and for each epoch condition, the model was evaluated with three different learning rates, which are 0.001, 0.05, and 0.01. The best precision of 99.5% was found with the following hyperparameters: epoch as 20, momentum = 0.9, and learning rate as 0.01. The results are shown in Table 3.

**Table 3.** Accuracy comparison with different epochs and learning rates.

| Epoch | 10 | 20 | 30 |
|---|---|---|---|
| **Learning Rate** | Accuracy | | |
| 0.001 | 98.8% | 99% | 99.2% |
| **0.01** | 98.3% | **99.5%** | 99.01% |
| 0.05 | 97% | 98.7% | 99.3% |

*4.1. Performance Analysis*

Figure 5a,b illustrate the training and validation accuracy/loss. At approximately the 9th iteration, the accuracy reached nearly 100%, as observed in Figure 5a, where the highest overall accuracy achieved throughout the testing phase is 99.6%. In Figure 5b, the curve starts to decline steeply at first, although there are some fluctuations because of the short batch size of 32. The fluctuations started to fade at approximately the 13th epoch and remained at almost zero.

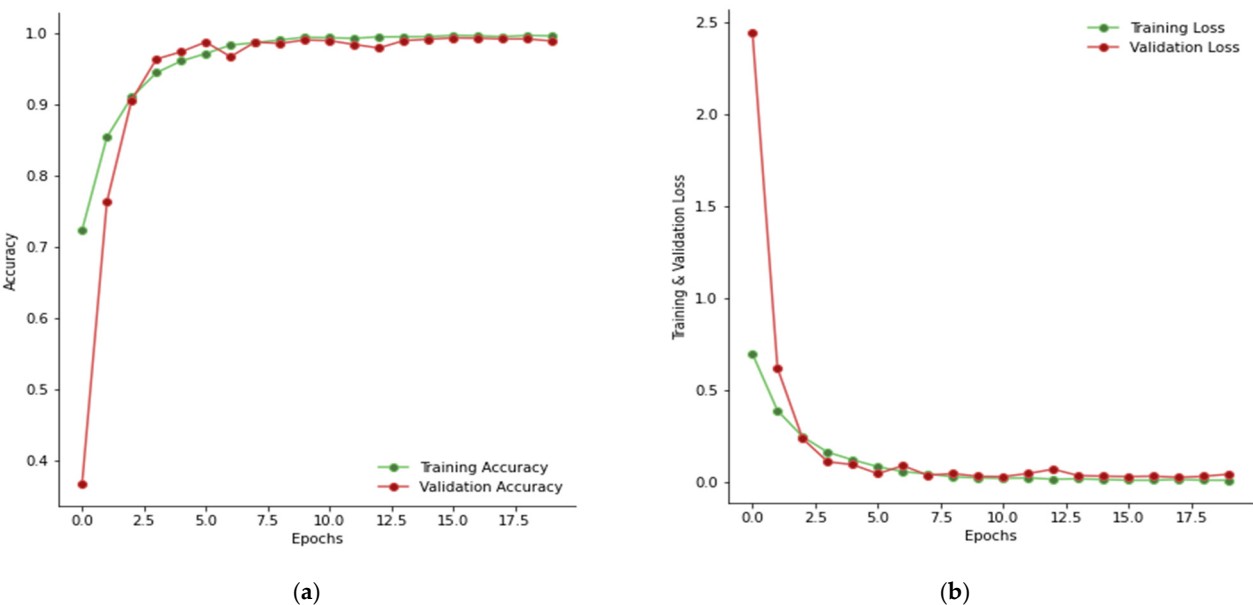

(**a**)                    (**b**)

**Figure 5.** (**a**) Epoch vs. training and validation accuracy, and (**b**) Epoch vs. training and validation loss.

### 4.2. Confusion Matrix

The confusion matrix that evaluates the system's performance is shown in Figure 6. The predicted values or system output are represented on the X axis, whereas the true labels or ground truth are represented on the Y axis. This is used to calculate evaluation metrics, such as precision, recall, sensitivity, specificity, and accuracy using (3), (4), (5), (6), and (7), respectively, which are used to evaluate the models' performance, such as [51].

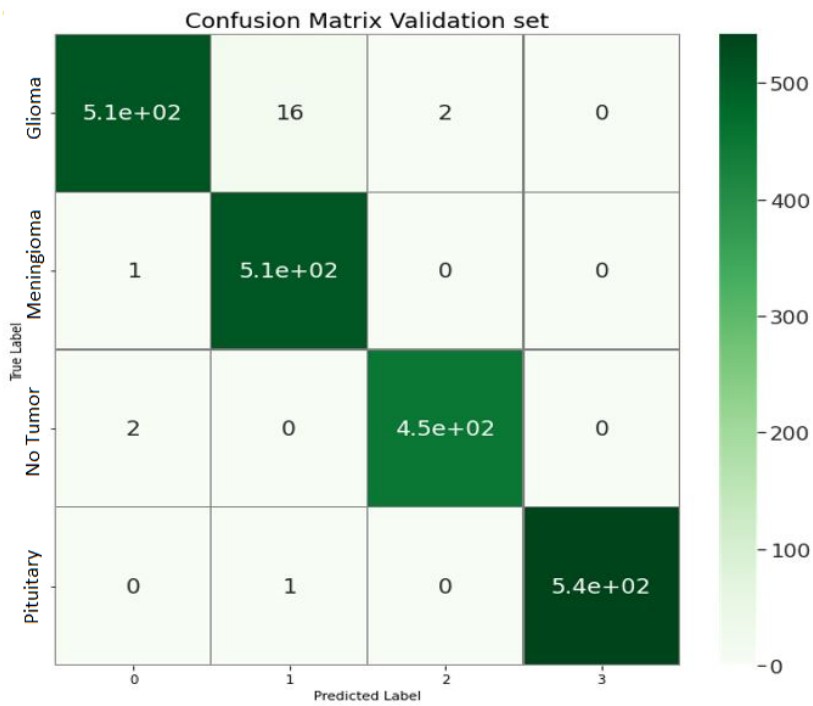

**Figure 6.** Confusion matrix of the validation dataset.

$$precision = \frac{tp}{tp + fp} \qquad (3)$$

$$sensitivity = \frac{tp}{tp + fn} \qquad (4)$$

$$specificity = \frac{tn}{tn + fn} \qquad (5)$$

$$accuracy = \frac{correct\ predictions}{all\ predictions} \qquad (6)$$

$$F_1\ Score = 2.\frac{precision.recall}{precision + recall} \qquad (7)$$

The confusion matrix, the F1 score, and the Mean Squared Error are used to evaluate the proposed system. Sensitivity, Specificity, Accuracy, Precision, and Recall are computed using True Positive or TP (True Positive), False Negative or FN (False Negative), True Negative or TN (True Negative), and False Positive or FP (False Positive). The best score for precision, recall, and specificity are the bold values in Table 4. An accuracy of 98.98% is achieved for Glioma, 99.13% for Meningioma, 99.95% for Pituitary, and 99.81% for No Tumor.

**Table 4.** Performance analysis of the proposed model.

| Tumors | TP | TN | FP | FN | Precision | Recall | Specificity | Accuracy | F1-Score |
|--------|----|----|----|----|-----------|--------|-------------|----------|----------|
| Glioma | 529 | 1510 | 3 | 18 | 0.994 | 0.967 | 0.988 | 98.98 | 0.98 |
| Meningi-oma | 511 | 1532 | 17 | 1 | 0.967 | **0.998** | **0.999** | 99.13 | 0.98 |
| No Tumor | 455 | 1601 | 2 | 2 | 0.996 | 0.996 | 0.998 | 99.81 | 0.99 |
| Pituitary | 543 | 1516 | 0 | 1 | **1.00** | **0.998** | **0.999** | 99.95 | 0.99 |

## 5. Discussion

The final architecture was formed by applying multiple parameters, including the changes in the dropout layers and optimization using SGD, to the VGG-16 configuration. Due to the multiple models and datasets used, it is difficult to compare the findings to other approaches reported at the state-of-the-art. However, we did manage to compare existing methods for detecting and classifying brain tumors in Table 5. We have applied the same dataset to our proposed model which is used in the following studies. The approach taken was segmentation-free.

**Table 5.** Comparison of the proposed structure with existing studies.

| Serial | Author | Model Used | Dataset Used | Model Accuracy | Our Model Accuracy |
|--------|--------|------------|--------------|----------------|--------------------|
| 1 | Paul et al. [52] | Fully Connected Network (FCN), CNN CapsNets incorporated with coarse tumor boundary | 3064 T1-weighted contrast-enhanced images with three kinds of brain tumor [44]. | 91.43% | 96.4% |
| 2 | Afshar et al. [53] | | 3064 T1-weighted contrast-enhanced images with three kinds of brain tumor [44]. | 90.89% | 96.4% |
| 3 | Anaraki et al. [54] | Genetic Algorithm (GA) | 3064 T1-weighted contrast-enhanced images with three kinds of brain tumor, combined with data from other sources | 94.2% | 95.3% |

The authors of [54] adopted a GA to identify the network architecture, although the prediction results did not implement the best one. In [53], the author incorporated the coarse tumor boundary as an additional input to assist the network to produce better outcomes. Furthermore, the authors of [52] applied only two convolutional layers, each having 64 kernels. Therefore, the proposed method predicts the best result when compared to

other relevant past research on multi-class classification types, demonstrating the proposed system's reliability.

Additionally, in Table 6, after comparing the proposed model with existing models, it is clear that our model provides the highest accuracy, while popular architectures, such as EfficientNet and ResNet remain behind. Even though our model is not pre-trained, it achieved 99.5% accuracy, whereas the pre-trained EfficientNet and ResNet models generated between 87–99.3% accuracy, which proves the validity of the modified VGG architecture.

**Table 6.** Comparison of the proposed structure with existing popular architectures.

| Model | Precision | Recall | F1-Score(macro) | Accuracy | Pre-Trained |
|---|---|---|---|---|---|
| EfficientNetB0 [55] | 0.942 | 0.941 | 0.941 | 0.941 | NO |
| EfficientNetB0 [55] | 0.993 | 0.993 | 0.993 | 0.993 | YES |
| Resnet50 [56] | 0.878 | 0.88 | 0.878 | 0.879 | YES |
| Resnet152 [56] | 0.889 | 0.885 | 0.885 | 0.885 | YES |
| VGG16 [48] | 0.980 | 0.980 | 0.980 | 0.980 | NO |
| **Modified-VGGNet** | 0.997 | 0.988 | 0.985 | **0.995** | NO |

As the proposed model is a modified version of the VGG-16 architecture, after comparing the precision, recall, F1-score and accuracy, it can be seen that the modified VGGNet is performing better than the base VGG-16 model in Table 7. The VGG-16 model has been trained using the same hyperparameters we have used in the modified VGGNet. We used learning rate = 0.01, momentum = 0.0 as hyperparameters for VGG-16 and modified VGGNet experiments and for other CNN architectures (ResNet, EfficientNet), we used learning rate = 0.001 and momentum = 0.9.

**Table 7.** Comparison of the proposed model with VGG-16.

| Tumors | Precision (VGG-16) | Precision (Proposed Model) | Recall (VGG-16) | Recall (Proposed Model) | F1-Score (VGG-16) | F1-Score (Proposed Model) | Accuracy (VGG-16) | Accuracy (Proposed Model) |
|---|---|---|---|---|---|---|---|---|
| Glioma | 0.98 | 0.994 | 0.95 | 0.967 | 0.97 | 0.98 | | |
| Meningioma | 0.97 | 0.967 | 0.98 | 0.998 | 0.97 | 0.98 | 98% | 99.5% |
| No Tumor | 0.98 | 0.996 | 1.00 | 0.996 | 0.99 | 0.99 | | |
| Pituitary | 0.99 | 1.00 | 0.99 | 0.998 | 0.99 | 0.99 | | |

Even though the total parameters of our model are much higher than the existing models, the evident advantage of having a large number of parameters is the ability to describe far more intricate functions than those with fewer parameters. Deep Neural Networks deal with a large number of training and testing parameters. The ability of neural networks to fit different sorts of information as the number of parameters rises is what makes them so powerful. Our model is versatile enough to describe the necessary mapping because it has numerous parameters. However, because of overfitting, this power is what causes the model to be weak. To avoid overfitting, models can always use more regularization. By utilizing batch normalization and max pooling, we were able to accommodate it. In our model, there are six blocks, and for each block, a conv2D layer followed by a max-pooling layer, and a batch normalization were placed. A conv2D layer's filter or kernel applies an elementwise multiplication to the 2D input data by "sliding" over it. It will therefore combine the outcomes into a single output pixel. The outcomes are then down-sampled or pooled feature maps that stress the patch's most prominent feature. The output can then be normalized in the same manner and distributed among the feature maps. The work was done by batch normalization. Instead of using the entire data set, it is done in mini-batches. It facilitates learning by accelerating training and utilizing higher

learning rates. Although it is slow because of this entire block, the approach appears to be more effective than others.

## 6. Conclusion and Future Work

Since we could achieve an accuracy of 99.5%, this system can indeed be used for industrial applications because of having such remarkable results. Compared to other research on multiclass classification, like SVM and KNN of [11], CNN of [52,53], and GA-CNN of [54], an accuracy of, respectively, 91.28%, 91.43%, 90.89%, and 94.2% were achieved. With their dataset, we obtained an accuracy of 95.7%, 96.4%, 96.01%, and 95.3%, respectively. However, with our modified VGG-16 and test set, we found a much better accuracy of 99.5%. The system is yet to be trained to detect brain tumors at an early stage. The identification is unquestionably crucial when it comes down to the health of a human being. If detection faces a complication and generates false results, it might be fatal to a person. One of the crucial limitations of this model is that it is comparatively slow as it contains high numbers of parameters, which can be solved by utilizing various other efficient existing models. Additionally, this model cannot detect the exact location of the tumor. The model can be trained to work with 3D images, which will further make it possible to locate the tumor's position. It can also be trained to detect brain tumors at an early stage.

**Author Contributions:** All authors contributed equally to this manuscript. All authors have read and agreed to the published version of the manuscript.

**Funding:** This research received no external funding.

**Institutional Review Board Statement:** Ethical review and approval were waived for this study due to the nature of it. The research does not use any sensitive personally identifiable data, forming no ethical issues.

**Informed Consent Statement:** Informed consent was obtained from all subjects involved in the study.

**Data Availability Statement:** Not applicable.

**Conflicts of Interest:** The authors declare no conflict of interest.

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
