# Peer review of "A CNN-Based Strategy to Classify MRI-Based Brain Tumors Using Deep Convolutional Network"

_applsci, doi:10.3390/app13010312_

Round 1

Reviewer 1 Report

 1. Mention the contributions of the study in the last paragraph of the introduction section

2.       In section 2 the paragraphs are very lengthy please shorten it and  need to include recently published works such as

DOI: https://doi.org/10.1007/s12652-022-04373-z

3.       Write the pseudocode of the proposed method

4.       Authors used evaluation metrics such as accuracy, sensitivity, and specificity; please cite such as DOI: https://doi.org/10.1016/j.eswa.2022.118045

5.      Mention the hyperparameters of the experimental results.

6.      Include the results of the original VGG and your Modified-VGGNet and how you modify it.

Compare your results with baseline methods such as

Doi:  10.1109/JBHI.2022.3171663

Author Response

Please see the attachment - Authors' Responses to Reviewer's Comments (Reviewer 1).

Reviewer 2 Report

The author addressed MRI based brain tumor classsification using modified VGG-16. Based on my review point, I have some questions and suggestions to improve the quality of the paper.

1. Please, correct the grammar mistake in the Abstract. It's a very unfortunate mistake that occurs at the beginning of the manuscript and gives the worst idea.

2. In Introduction section is not clear with the research contributions and highlights. I suggest the author include the contributions and motivations of the work in the introduction section. 

3. In Figure 1, specified preprocessing process with coloring, orienting, resizing but in the section 3.2, it was not discussed.

4.Section 3.3 describes the proposed method, but the description is not clearly mentioned how the generic VGG is improved. 

5. Figure 4 was not clear, and the representation for traning loss and training accuracy is mismatching. 

6. How modified VGGNet improves its accuracy than CNN model? Various literature study has addressed CNN model and provided better efficacy. Justify, the outcome of your model with CNN model. (for Eg: https://www.sciencedirect.com/science/article/pii/S277252862200022X#se0130)

7. The conclusion section needs to be refined with importance of your study and future directions.

8. Clean up the grammar and English mistakes in the entire manuscript.

Author Response

Please see the attachment - Authors' Responses to Reviewer's Comments (Reviewer 2).

Reviewer 3 Report

The paper presented a brain tumor classification method based on deep learning. However, the novelty of the proposed method is less significant. Furthermore, the presentation of the method is traditional. The impact of the results based on splitting cannot be generalized on a single dataset.  Finally, the quality and novelty are not suitable for this journal. I hope the best for the authors in the next time. Thanks

Author Response

Please see the attachment - Authors' Responses to Reviewer's Comments (Reviewer 3).

Reviewer 4 Report

This paper “An Advanced Strategy to Classify MRI-Based Brain Tumors Using Deep Convolutional Network” aimed to create a fast and reliable system, which can help medical professionals in identifying brain tumours. The authors claimed to establish a much more efficient and error-free classification method, which is trained with a comparatively substantial number of real datasets rather than an augmented one. Authors also claimed that using a modified VGG-16 (Visual Geometry Group) architecture on 10,153 MRI (Magnetic Resonance Imaging) images with 3 different classes of brain tumours, the network performs significantly well, with an overall accuracy of 99.5%.

The topic is justified. The paper could be further improved if the following remarks are taken into consideration:

1.       The title claimed an “An Advanced Strategy to Classify MRI-Based Brain Tumors Using Deep Convolutional Network”, but in the conducted research, authors used VGG-16, which is a CNN-based model. Why authors claimed it is “An Advanced Strategy”?

2.       ABSTRACT: The text should include more details about the proposed methodology, numerical results achieved (precision, recall, sensitivity, specificity, and accuracy), and a comparison with other state-of-the-art methods.

3.       Several grammatical mistakes were found in the whole draft of the article; the authors need to fix these.

4.       Introduction section lacks a proper introduction of the whole of the conducted research, background, justification of the research, and major contributions of the study. The contribution may be key fold in the introduction section.

5.       Add more recent related (conducted in the year 2021-2022), studies in related works, currently the support of most recent state-of-the-art studies is missing in the article.

6.       The caption of Table 1 should be “Some other state-of-artwork.”->” Some other state-of-the-artwork.”, the word “the” is missing, as mentioned in line no. 135 also.

7.       Better to add evaluation measures too in Figure 1.

8.       Dataset: the volume of the dataset is ok with these experiments.

9.       Pre-processing: what was the original size of the dataset of different frames, which was resized to 200x200x1? Did not the authors think resizing the medical slices (MRI images in this case), can lose valuable information (texture in this case), which could affect overall classification results? Did the authors verify this impact?

10.   The authors divided the dataset into 80% for training and 20% for testing, why not for validation?

11.   Line no. 167 states, “Here, we followed the Visual Geometry Group (VGG) architecture.”, but I found a proposed CNN model in figure 3. What is borrowed from VGG in the proposed CNN model?

12.   First part of figure 3 is labelled as a convolutional layer, while it has other layers too (activation functions and normalization layers), so this could not be the only convolution, similarly, the second part of figure 3 is labelled as a dense layer, while it has activation function too. I would suggest to re-think while labelling the figure, suitable to their functionality (combined).

13.   Legends (captions/labels) of most of the figures are too short, which could not elaborate on the figure (s).

14.   Confusion matrix shown in figure 6, must show classes (Glioma, Meningioma, Pituitary, and No Tumor) rather than simply 0, 1, 2, and 3.

15.   As per the layered architecture of the proposed model shown in Table 2, the total parameters of the proposed CNN model are too high as compared to the other pre-trained CNN models (squeezenet, mobilenetv2, shufflenet, efficientnetb0, nasnetmobile, googlenet, resnet18, resnet50, xception, darknet19, etc.). How are the authors claiming the proposed model is fast and advanced?

16.   Why VGG-16 is the choice for follow? Despite this, several other CNN-based state-of-the-art models are there to use in any fashion, which are very efficient in their performance even.

17.   Regarding the comparison of the results (Table 5), based on what the comparison is made between the proposed and state-of-the-art studies? Did state-of-the-art studies also use the same dataset?

18.   The authors compared their results with references no, 48, 49, and 50, which are conducted in the year 2017, 2019, and 2019 respectively. These are not most recently conducted related.

19.   The motivation is not clear. Please specify the importance of the proposed solution.

20.   Discuss the limitations of the proposed method with their possible solutions in the future work section.

21.   Conclusion section should be ‘Conclusion and Future Work’ as it contains future work-related information too.

Author Response

Please see the attachment - Authors' Responses to Reviewer's Comments (Reviewer 4).

Round 2

Reviewer 2 Report

Dear authors,

Thank you for addressing all my comments

Reviewer 3 Report

Thanks for the author's effort in the paper. However, the contribution is too little to be published in this journal. 

Reviewer 4 Report

The authors addressed my comments convincingly.